# Analysis of the Relationship among Cognitive Impairment, Nutritional Indexes and the Clinical Course among COVID-19 Patients Discharged from Hospital—Preliminary Report

**DOI:** 10.3390/nu14081580

**Published:** 2022-04-11

**Authors:** Jakub Klimkiewicz, Daniel Pankowski, Kinga Wytrychiewicz-Pankowska, Anna Klimkiewicz, Paulina Siwik, Joanna Klimczuk, Arkadiusz Lubas

**Affiliations:** 1Department of Anesthesiology and Intensive Care, COVID-19 ICU, Military Institute of Medicine, 04-141 Warsaw, Poland; jklimkiewicz@wim.mil.pl; 2Faculty of Psychology, University of Economics and Human Sciences in Warsaw, 01-043 Warsaw, Poland; d.pankowski87@gmail.com; 3Faculty of Psychology, University of Warsaw, 00-183 Warsaw, Poland; 4Department of Psychiatry, Medical University of Warsaw, 02-091 Warsaw, Poland; 5Department of Ophthalmology, Central Clinical Hospital of the Ministry of the Interior and Administration in Warsaw, 02-507 Warsaw, Poland; paulina.siwik@cskmswia.gov.pl; 6Department of Neurology, Military Institute of Medicine, 01-755 Warsaw, Poland; jklimczuk@wim.mil.pl; 7Department of Internal Diseases Nephrology and Dialysis, COVID-19 Department, Military Institute of Medicine, 01-755 Warsaw, Poland; alubas@wim.mil.pl

**Keywords:** COVID-19, cognitive impairment, nutrition, phosphate, albumin

## Abstract

Numerous data indicate the presence of cognitive impairment in people who have undergone COVID-19, often called COVID Fog (CF). This phenomenon persists even 6 months after infection, and its etiology and pathogenesis are not fully known. The aim of this article was to analyze the relationship among cognitive functioning, clinical data and nutrition indexes in patients discharged from the COVID-19 hospital of the Military Institute of Medicine, Warsaw, Poland. The sample comprised 17 individuals—10 women and 7 men, with ages of 65 ± 14 years. Cognitive impairment was measured with the use of the Montreal Cognitive Assessment (MoCA). The nutrition parameters included: hemoglobin, red blood cells, total cholesterol and its fractions, triglycerides, total protein, albumin, urea, creatinine, phosphates, calcium and sodium. The analysis showed that albumin concentration significantly correlated with the total MoCA score and especially with the short-term memory test score. Conversely, total cholesterol, and especially LDL concentrations, were highly and negatively associated with the MoCA score. In conclusion: markers of nutritional status are correlated with the severity of CF. Individuals with malnutrition or risk of malnutrition should be screened for CF. Further studies need to be performed in this area.

## 1. Introduction

Cognitive dysfunction in the group of people with medical conditions is a growing clinical problem [1,2]. An increasing number of analyses focus on biological pathomechanisms [3], factors related to treatment [4] and psychological correlates [5] with this dysfunction. Cognitive impairment may lead to difficulties in everyday functioning, lowering the quality of life and promoting depressive symptoms [6].

Recent studies indicate the presence of COVID-19 neuropsychiatric complications, such as depression [7], delirious states [8] or cognitive decline [9]. Cognitive impairment related to COVID-19 is also a symptom that is part of the so-called “long COVID-19”. Long COVID-19 is described to have, among others, symptoms of fatigue, headache, dyspnea, anosmia, phantosmia and phantogeusia. Its risk factors consist of higher age, higher body mass index and female gender [10,11].

COVID-19 is associated with a high risk of complications from the central nervous system (CNS). Cognitive disorders of memory, attention or executive functions among COVID-19 convalescents is often called the brain fog (COVID Fog—CF). CF leads to psychomotor deterioration and chronic fatigue syndrome, resulting in poor functioning and lowering quality of life. CF may affect up to 81% of patients after COVID-19; Graham et al. [12], in a prospective study on a group of 100 patients, noticed that non-hospitalized COVID-19 “long haulers” experience prominent and persistent CF and fatigue that affect their cognition and quality of life. In the studied group, the most common findings were also: headache (68%), numbness/tingling (60%), dysgeusia (59%), anosmia (55%) and myalgias (55%). Prevalence of CF may be even greater among patients with severe forms of COVID-19. The preliminary assessment found that 83% of COVID-19 inpatients had at least mild cognitive impairment [13]. Ferrucci et al. [14], in an analysis of patients 5 months after hospital discharge, noted that 42.1% had processing speed deficits, 26.3% showed delayed verbal recall deficits and 21% presented with deficits in both processing speed and verbal memory. Moreover, infection with Severe Acute Respiratory Syndrome Coronavirus 2 (SARS-CoV-2) was found to correlate with a higher incidence of depression and anxiety disorders [15].

The literature indicates the occurrence of various neurological complications related to SARS-CoV-2 infection itself, such as encephalopathies, and acute disseminated encephalomyelitis with hemorrhages and necrotic change, as well as inflammatory changes associated with subsequent complications, e.g., post-infectious inflammation and cytokine-related hyper inflammation [16]. All the abovementioned factors may explain the occurrence of lesions in the central nervous system. The exact mechanism of these changes is not fully known, but researchers suggest two potential pathways in which infection may affect CNS: direct and indirect [17]. What was likely the first evidence linking SARS-CoV-2 infection and CNS was presented by Wang et al. [18], who noted viral RNA in the cerebrospinal fluid. One of the probable mechanisms of brain lesions and CF is SARS-CoV-2 neuroinvasion. Viral spike protein (“S”—spike protein) has the ability to induce local inflammation by binding angiotensin II-converting-enzyme receptors on endothelial cells of small brain vessels. This predisposes them to the damage of the vascular wall, formation of microclots and consequently cerebral ischemia. In addition, the S protein has the ability to increase the blood–brain barrier permeability and to infect neurons and glial cells [19]. Another route of neuroinvasion is trans-synaptic and axonal virus transportation through the olfactory and optic nerves to the CNS [20]. Additionally, the “Trojan horse” mechanism using immune system cells may explain neuroinvasion caused by SARS-CoV-2 [21]. Increased permeability of the blood–brain barrier also enables direct neuroinvasion, which promotes further unsealing of the vascular endothelium and additionally increases the risk of blood clots and stroke. On the other hand, the indirect influence of COVID-19 on CNS function is characterized by systemic changes caused by COVID-19, such as, e.g., hypoxia, which may also lead to cognitive dysfunction [22].

Nuclear Magnetic Resonance (NMR) studies in patients with CF showed the presence of changes in the temporal lobes, multifocal changes in white matter and micro-hemorrhages [23]. Positron emission tomography and computed tomography (PET-CT) scans revealed reduced metabolism in the area of the frontal lobes. The prefrontal cortex located in the frontal lobes is responsible for higher neurological functions, i.e., awareness, thinking, association, behavior and memory [24]. Damage in these areas corresponds with symptoms of CF. Studies of transcranial magnetic stimulation in patients after COVID-19 revealed the disorders of short-term and long-term cortical inhibition associated with the function of receptors for gamma-aminobutyric acid (GABA-A and GABA-B) [25]. This suggests the involvement of the GABA messenger system in the pathogenesis of CF. Additionally, hypoxia may be pivotal in the development of CF. Hypoxia induces and enhances pre-existing inflammatory processes. High concentrations of pro-inflammatory cytokines, especially interleukin 1 and interleukin 6, are associated with deterioration of the central nervous system function [26]. The persistence of neuropsychological changes after cessation of systemic inflammation may also indicate activation of destructive autoimmune mechanisms in CNS by COVID-19. The next mechanism promoting CNS dysfunction is SARS-CoV-2 multiplication inside mitochondria. This leads to an intracellular energy deficiency secondary to a lack of ATP and disrupts neurotransmission, promoting demyelination and death of the nerve cells. This is why COVID-19 may also be perceived as a neurodegenerative disease [27]. Apart from pathophysiological mechanisms, COVID-19 may trigger deterioration of CNS function due to long hospitalization, stress and sleep deprivation. Of course, psychological and sociological consequences are a burden for every hospitalization, but COVID-19 treatment is long and usually connected with isolation from relatives and the anonymity of medics due to the use of protective equipment. One of the potential factors that may also affect cognitive functioning is the level of nutrition [28].

There is evidence linking nutrition status with the clinical course of COVID-19. James and associates divided the field of nutrition into 13 nutrition-related components and their potential interactions with COVID-19. These components are overweight, obesity and diabetes; protein-energy malnutrition; anemia; vitamins A, C, D and E; PUFAs; iron; selenium; zinc; antioxidants; and nutritional support [29]. Nutrition is an important player when comes to susceptibility to infection with SARS-CoV-2 and progression of COVID-19. Diet affects mucosal immune function, immune cells’ function, viral replication and finally inflammation [29]. As nutrition is important for both COVID-19 progression and cognitive impairment, we hypothesize that malnutrition could trigger CF symptoms, and conversely, proper nutrition is able to ameliorate cognitive dysfunction after COVID-19. As malnutrition is widespread among both surgical and medical patients [30,31], we found it useful to investigate the potential correlation between nutrition states and CF.

## 2. Materials and Methods

The study was approved by Bioethics Committee of Military Institute of Medicine. Study took place during summer of 2021 in temporary COVID-19 hospital located in Military Institute of Medicine, Warsaw, Poland. Hospital comprised both general ward and intensive care unit. The study included patients with no previous history of cognitive impairment, discharged from the COVID-19 hospital after severe COVID-19. Specific inclusion criteria were age of above 18 years and being discharged from hospital after severe form of COVID-19. Exclusion criteria were known previous cognitive impairment or cerebrovascular incident, previous head injury, stroke or transient ischemic attack during present hospitalization, factors preventing individuals from proper filling of questionnaire (e.g., sight or hearing impairment, tracheotomy, muscle weakness due to polyneuropathy). Additionally, patients discharged to a nursing home or other ward for further treatment were excluded from the study. Individuals who agreed to participate in the study, and gave informed consent, filled in the MoCA v 7.2 questionnaire in the Polish adaptation when discharged from hospital [32]. The MoCA scale includes tests assessing cognitive abilities in the form of visual–spatial and executive functions (maximum 5 points), naming skills (3 points), short-term memory (5 points), attention and its selectivity (6 points), language functions (3 points), abstract thinking (2 points) and allopsychic orientation (6 points). Maximal achievement is 30 points, and the standard threshold of cognitive impairment is the value of ≤26 points. For cognitive functions estimated in more than one test, the sum of specific results for the given cognitive ability tests was divided by the sum of their maximal scores to achieve a decimal fraction ranging from 0.00 to 1.00, which was considered for statistics. Visuospatial function expresses the results of two tests: the drawing figure and joining points test and the clock drawing test. Attention function consisted of attention digits, letters and subtraction tests. Language function express the results of the repetition and the fluency tests.

In order to assess the severity of COVID-19 infection, the following data from hospitalization were retrospectively assessed: the lowest peripheral hemoglobin saturation with oxygen (SatO_2_), the severity of inflammation measured by the maximum concentration of C-reactive protein (CRP), the length of hospitalization in days and the intensity of oxygen therapy. In order to assess the intensity of respiratory support with oxygen therapy (scale 1–5), the following criteria were adopted: 1—nasal cannula oxygen therapy (up to 5 L/min); 2—face mask (without and with a reservoir; up to 10 L/min); 3—face mask with reservoir (>10 L/min); 4—mask with a reservoir and nasal cannula oxygen therapy (>30 L/min); and 5—high-flow oxygen therapy, non-invasive ventilation (NIV) and invasive mechanical ventilation.

In order to assess the nutritional status, the results of blood serum tests in the first days of hospitalization were retrospectively collected for: sodium (136–145 mmol/L); total protein (6.4–8.3 g/dL); albumin (3.9–4.9 g/dL); total cholesterol (120–200 mg/dL); LDL (50–130 mg/dL); HDL (35–65 mg/dL); triglycerides (35–165 mg/dL); calcium (8.6–10.2 mg/dL); phosphates (2.6–4.5 mg/dL); creatinine (0.7–1.2 mg/dL); urea (18–55 mg/dL); erythrocytes (4.0–10.0 m/mL); and hemoglobin (11.0–18.0 g/dL). We chose those parameters as possibly contributing to cognitive status and connected with lipid metabolism (total cholesterol and its fractions), nitrogen equilibrium (albumin, total protein, creatinine, urea) and feeding status (phosphates, calcium, albumin, lipids, erythrocytes and hemoglobin). Plasma sodium was used as a monitor of volume status and to exclude hyponatremia as a trigger for cognitive impairment.

### Statistical Analysis

The results are presented in the form of the mean with standard deviation and the median with extreme values. The compliance of the variable distribution with the normal distribution was checked using the Shapiro–Wilk test. Correlation analysis was performed using Pearson’s test for variables with a distribution close to the normal value; otherwise, the Spearman’s test was used. In order to determine the relationship between the analyzed variables, partial correlations were made, in which the age of the respondents and the level of education were controlled. Due to the small size of the sample, calculated partial correlation coefficients were interpreted based on the significance level (*p*-value) and the effect size (small-low for │r│ ≤ 0.24, medium-moderate for │r│ > 0.24 and large-high for │r│ > 0.37) [33]. For the sample size calculation, analysis of correlation between short-term memory score and albumin concentration (r = 0.772; *p* = 0.072) was performed in the group of 10 initially included patients. To achieve the power of 0.90 and *p* < 0.05, minimal sample size was 12. For each test, a level of two-tailed *p* < 0.05 was considered significant. Statistical analysis was performed with the use of Statistica 12 software (StatSoft Inc., Cracow, Poland).

## 3. Results

A total of 17 patients (10 women, 7 men; age 65.0 ± 14.0) were enrolled in the study. The mean length of hospitalization was 20.2 ± 15.7 days (median 13; range 5–63); mean maximum C-reactive protein (CRP) concentration was 10.26 ± 8.5 mg/dL; mean lowest peripheral blood saturation was found to be 86.5 ± 6.73%; and the average degree of oxygen therapy was 1.9 ± 1.5 (median 1.0; range 0.0–4.0). The mean MoCA score was 21.59 ± 5.23 points. Only five patients had a normal MoCA score. Results of the categories assessed with MoCA and results of disease severity and nutrition parameters are presented in Table 1.

First, partial correlations between clinical factors and the overall result and MoCA subscales were calculated, controlling for the age and education of the respondents. The results of the analyses are presented in Table 2.

The results presented in Table 2 show that the magnitude of the effect of the strength of the relationship between the analyzed variables ranged from no effect (e.g., the relationship between the subtest performance clock and the length of hospitalization or the MoCA total score with CRP max) to a large effect (e.g., the relationship between attention—subtraction and the duration of hospitalization). The correlation between the attention—letters score and respiratory support was statistically significant.

Subsequently, the relationship between the nutritional indexes, the performance level of the MoCA subscales and the overall result was determined (Table 3). Again, the analyses controlled for the age and education of the respondents. The analysis of the relationships among nutritional indices and subscale scores and the MoCA total score showed that the size of the effects of the relationship between the variables ranged from no effect (short-term memory and urea) through low and moderate to large effect size (for example, short-term memory and albumin, total cholesterol and LDL). The MoCA total score was highly associated with albumin, total cholesterol and LDL. Moreover, the correlation of the MoCA total score and especially short-term memory test score with albumin was statistically significant.

## 4. Discussion

In the presented study, the relationship between cognitive functioning factors assessed using neuropsychological screening methods, nutrition markers and the clinical course of COVID-19 was analyzed.

The conducted analyses showed that the magnitude of the effect of the strength of the relationship between the clinical indicators and the MoCA results is the highest in the case of repetition and SpO_2_ min; attention—subtraction and length of hospitalization; CRP and attention–digits and letters. The results of studies conducted in groups of chronically ill people confirm such observations—e.g., Hoth et al. [34]. Hoth and colleagues noted that high CRP levels are associated with worsening attention–executive–psychomotor performance. In other studies, statistically significant relationships between reaction time and CRP were found [35]. Additionally, the length of hospitalization in earlier studies was associated with the occurrence of cognitive difficulties; the factors that mediated this relationship included, inter alia, used treatment, stress and intensification of depressive symptoms [36]. Researchers also noticed that the deterioration of cognitive functioning occurs not only in people treated in critical care. In our study, the attention—letters test score correlated significantly with respiratory support—this could probably be explained by better oxygenation status. The different values of the correlation (and hence the size of the effect) may indicate a different range of influence of clinical variables on individual cognitive modalities; however, this thesis should be confirmed in longitudinal studies. It should also be noted that the correlations included the subscales of the screening tool; to better analyze the above relationships, the results should be confirmed using more accurate neuropsychological methods.

On the other hand, a greater number of significant correlations were found in the case of nutritional indices; selected MoCA subscales positively correlated with, e.g., the level of serum albumin and sodium.

In the case of albumin, the obtained results are consistent with other data from studies in people 65+, which indicated that a low albumin concentration is a risk factor for cognitive impairment [37,38]. In our study, lower albumin concentration correlated significantly with, e.g., worse short-term memory function and attention. On the other hand, cholesterol concentrations turned out to be negatively associated with most of the MoCA cognitive abilities. Scientific data indicate that high levels pose a greater risk of cognitive decline or dementia, especially in mid-life [39]. In this respect, our results are consistent with the observations of other researchers. Additionally, it was found that calcium and HDL levels may have played a protective role in most subscales. These data are consistent with the results of other studies which found that HDL levels were positively related to both the level of cognitive functioning assessed using screening and advanced neuropsychological tests [40,41]. Conversely, Bakeberg et al. noticed that a high level of HDL presents a sex-specific biomarker for cognitive impairment in females with Parkinson’s disease [42]. In turn, calcium-dependent signals are key triggers of the molecular mechanisms underlying learning and memory [43]. However, data from the literature show that higher concentrations of calcium levels are associated with a faster decline in cognitive function in late life (in people aged 75+) [44]. Contrarily, a study in Japan found that lower serum calcium may be associated with an increased risk of mild cognitive impairment conversion to Alzheimer’s disease [45].

In our study, phosphate concentration was negatively and highly correlated with most MoCA subscales as well as with the MoCA total score. The presented findings could be the result of a high-normal concentration of phosphates (3.65 ± 0.66 mg/dL) in the included patients. Malnutrition and re-alimentation frequently result in refeeding syndrome, in which swings of various biochemical serum parameters such as potassium, calcium, magnesium, zinc and phosphates are observed. Typically, in fully developed refeeding syndrome, hypophosphatemia is observed [46]. Also, hypophosphatemia ad admission is an independent risk factor for death among COVID-19 patients [47].

The presented study also had several limitations that may reduce the possibility of a wider generalization of results to the population of people with COVID-19. In our study, only patients infected with a virus strain were analyzed, which, on the one hand, may translate into a limited possibility of generalizing the results to all subjects who have undergone COVID-19, and on the other hand, it can be considered as a strong advantage of collected data. Thanks to this solution, the characteristics of people who have undergone COVID-19 were very homogeneous—any differences in individual results derived from factors other than the form of the virus. Another limitation of the study was the limited sample size. Therefore, many correlations, the coefficients of which were quite high, turned out to be statistically insignificant. A larger sample would increase the representativeness of the results, but due to the fact that the overarching goal was to preserve the homogeneity of the studied population, we decided not to include further patients in the study with a different variant of the virus. Another limitation is the cross-sectional nature of the study with regard to the MoCA results. On the basis of the collected data, it is not possible to determine whether the deficits in the field of cognitive functioning appeared after undergoing COVID-19 or were present before the infection. Additionally, the MoCA is a screening tool for the assessment of cognitive functions, which may be, for example, an indication for further, more in-depth analysis of cognitive functions. In subsequent studies, it is recommended that researchers include more extensive neuropsychological tests in the study, while people who report a subjective cognitive decline or an MoCA score below the cut-off point would indicate the need for a more detailed examination.

The above limitations also determine the potential further directions of research. Certainly, the next stage should be analyses carried out on a much larger sample, using, in addition, more advanced methods of neuropsychological assessment. In future research, it is also worth analyzing the cerebrospinal fluid or using neuroimaging methods, such as NMR and PET. In addition to clinical indicators, it is also worth taking into account other variables that may potentially be responsible for the development of cognitive decline, such as the severity of depressive symptoms or the level of fatigue. Due to the fact that the symptoms associated with the deterioration of cognitive functions are described by patients even many months after leaving the hospital, a longitudinal study that would identify the role of individual factors in the long-term persistence of cognitive dysfunctions and identify protective factors seems justified.

The obtained results are of great value for implementation in clinical practice. Thanks to the results of the analyses, it will be possible to select people who should participate in neuropsychological rehabilitation or other activities that may positively affect cognitive functions. As shown, cognitive dysfunctions may lead to a reduction in the quality of life, difficulties in returning to daily functioning, performing professional duties, etc. [48]. Early initiation of activities with a potentially beneficial effect could reduce the perceived negative consequences in the long term in the group of patients leaving hospital wards.

## 5. Conclusions

Nutrition factors correlate with the occurrence and severity of symptoms of CF. This may indicate that malnutrition could play a substantial role in the development of CF. Better nutritional status could be the key to ameliorating CF symptoms. However, further studies are needed to investigate and confirm that phenomenon.

## Figures and Tables

**Table 1 nutrients-14-01580-t001:** Results of the categories assessed with MoCA, disease severity and nutrition parameters.

Variable	Mean	Standard Deviation	Median	Minimum	Maximum
Drawing figure and joining points test	0.88	0.86	1.00	0.00	2.00
Clock drawing test	2.35	0.70	2.00	1.00	3.00
Visuospatial function *	0.65	0.26	0.60	0.20	1.00
Naming skills	2.53	0.62	3.00	1.00	3.00
Attention—digits	1.41	0.62	1.00	0.00	2.00
Attention—letters	0.71	0.47	1.00	0.00	1.00
Attention—subtraction	2.06	1.09	3.00	0.00	3.00
Attention function *	0.70	0.25	0.67	0.17	1.00
Repetition	1.59	0.62	2.00	0.00	2.00
Fluency	0.65	0.49	1.00	0.00	1.00
Language function *	0.75	0.32	1.00	0.00	1.00
Abstraction	1.35	0.79	2.00	0.00	2.00
Short-term memory	1.82	1.13	2.00	0.00	3.00
Allopsychic orientation	5.76	0.56	6.00	4.00	6.00
MoCA total score	21.59	5.23	21.00	10.00	29.00
Age (years)	65.00	14.01	63.00	43.00	86.00
Hospitalization length (days)	20.18	15.71	13.00	5.00	63.00
Highest CRP (mg/dL)	10.26	8.50	10.90	0.30	27.30
Lowest SatO_2_ (%)	86.47	6.73	87.00	75.00	97.00
Respiratory support (1–5)	1.88	1.45	1.00	0.00	4.00
Total protein (g/dL)	5.80	0.73	5.85	4.90	6.60
Albumin (g/dL)	3.58	0.73	3.60	2.50	4.30
Total cholesterol (mg/dL)	104.80	16.63	106.00	84.00	125.00
LDL cholesterol (mg/dL)	65.43	22.68	57.00	42.00	112.00
HDL cholesterol (mg/dL)	34.29	10.18	30.00	27.00	56.00
Triglycerides (mg/dL)	111.50	39.97	105.50	72.00	187.00
Calcium (mg/dL)	8.29	0.57	8.40	7.30	8.90
Phosphate (mg/dL)	3.65	0.66	3.55	3.10	4.40
Urea (mg/dL)	38.38	17.92	36.00	18.00	68.00
Creatinine (mg/dL)	0.98	0.24	0.95	0.60	1.40
Sodium (mmol/L)	137.75	1.83	137.50	135.00	140.00
Red blood count (mln/mcL)	4.21	0.71	4.29	2.61	5.33
Hemoglobin (g/dL)	12.25	2.10	12.00	8.00	15.30

* The sum of results of the specified function tests divided by the sum of maximal scores of these tests.

**Table 2 nutrients-14-01580-t002:** Partial Pearson’s correlations among MoCA tests, functions and the clinical parameters with age and the education level adjustment.

Test/Function	Hospitalization Length	CRP Max.	SpO_2_ Min.	Respiratory Support
Drawing figure and joining points test	0.08	−0.26	0.34	−0.17
Clock drawing test	−0.04	−0.31	0.22	0.01
Naming test	−0.20	0.10	0.06	−0.07
Attention—digits test	−0.15	**0.40**	−0.06	0.08
Attention—letters test	0.13	**0.40**	−0.19	**0.55 ***
Attention—subtraction test	**0.43**	−0.08	−0.20	0.17
Repetition test	−0.13	−0.13	**0.49**	−0.09
Fluency test	−0.08	0.23	0.15	0.19
Abstraction test	0.10	−0.17	0.26	−0.20
Short-term memory test	0.06	0.05	−0.08	0.11
Orientation test	0.36	0.25	−0.28	−0.01
Visuospatial functions	−0.08	−0.25	**0.39**	−0.16
Attention functions	0.32	0.28	−0.28	**0.41**
Language functions	−0.11	−0.07	**0.40**	0.08
MoCA total score	0.13	0.00	0.12	0.06

Bolded results—large effect size correlations; * *p* < 0.05.

**Table 3 nutrients-14-01580-t003:** Partial Pearson’s correlations between MoCA results and nutritional indicators with respect to age and the education level.

Test/Function	RBC	HGB	Sod	Calc	Phos	Crea	Urea	TProt	Alb	Chol	HDL	LDL	Tgc
Drawing figure and joining points test	−0.23	−0.05	0.22	−0.15	**−0.45**	0.18	0.17	−0.19	0.14	**−0.52**	−0.00	**−0.43**	0.32
Clock drawing test	**−0.54 ***	**−0.48**	**0.59 ***	−0.21	−0.11	−0.21	−0.07	−0.05	−0.10	−0.03	**−0.46**	**0.53**	0.23
Naming test	−0.17	−0.14	0.27	−0.03	0.16	0.03	−0.21	0.22	0.14	**0.58**	0.19	0.05	**0.56**
Attention—digits test	−0.09	0.12	**0.44**	0.33	−0.07	**−0.61 ***	**−0.38**	−0.01	−0.20	0.17	0.17	−0.15	**−0.41**
Attention—letters test	0.03	0.24	0.21	**−0.68 ***	**−0.50**	−0.32	−0.17	**−0.75 ***	**−0.71 ***	−0.14	0.23	−0.13	**−0.80 ***
Attention—subtraction test	−0.33	−0.35	0.21	−0.09	−0.20	0.04	0.04	0.14	**−0.41**	**−0.59**	0.19	**−0.48**	−0.01
Repetition test	−0.05	0.06	−0.15	0.14	**−0.54**	0.15	**0.47**	−0.08	0.22	−0.25	0.00	−0.32	0.35
Fluency test	−0.12	0.02	0.21	0.06	**−0.60**	−0.04	0.29	0.01	0.25	**−0.63**	−0.23	**−0.53**	0.05
Abstraction test	0.24	0.25	−0.13	0.15	**−0.54**	0.15	0.20	0.10	**0.42**	**−0.47**	0.09	**−0.58**	0.21
Short-term memory test	0.41	0.32	0.11	0.15	**−0.48**	0.12	0.00	0.32	**0.73 ***	**−0.62**	0.14	**−0.95 ***	−0.14
Orientation test	−0.40	**−0.44**	0.33	0.17	−0.15	−0.11	−0.18	0.01	−0.02	0.23	0.19	0.05	**0.40**
Visuospatial functions	**−0.45**	−0.36	**0.41**	−0.31	−0.29	0.05	0.30	−0.14	0.05	**−0.41**	**−0.71**	0.23	0.30
Attention functions	−0.33	−0.17	**0.53 ***	−0.31	**−0.54**	**0.49**	0.14	−0.23	−0.04	**−0.64**	0.29	**−0.79**	**−0.65**
Language functions	−0.06	0.09	0.04	−0.13	**−0.57**	0.07	0.28	−0.05	0.29	**−0.38**	**0.40**	0.31	0.09
MoCA total score	−0.16	−0.06	0.36	−0.04	**−0.60**	−0.04	0.05	0.03	**0.47 ***	**−0.60**	0.12	**−0.74**	0.14

Sod—sodium; Calc—calcium; Phos—phosphates; Crea—creatinine; TProt—total protein; Alb—albumin; Chol—total cholesterol; bolded results—mark large effect size correlations; * *p* < 0.05.

## Data Availability

The dataset is with the authors and available on request.

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
