# Peer review of "Analysis of the Relationship among Cognitive Impairment, Nutritional Indexes and the Clinical Course among COVID-19 Patients Discharged from Hospital—Preliminary Report"

_nutrients, 2022, doi:10.3390/nu14081580_

Round 1

Reviewer 1 Report

This manuscript provides data on the relationship between cognitive impairment, some nutritional indices, and the clinical course among Covid-19 patients discharged from the hospital. Overall, I think this manuscript can be a valuable contribution to these actual and important topics, but it needs some corrections before publishing.

  • The Abstract lacks the conclusion sentence or future directions.
  • Please use the “COVID-19 “abbreviation consistent throughout the whole manuscript.
  • What means “MRI studies…”? (Page 2, line 85)
  • The authors should provide more data related to the study design in the Materials and Methods section. Where was a study conducted? When? Add inclusion criteria, followed by exclusion criteria, and then provide data on the methodology. Unify the name of the Ethical Committee in the Material and method section (Page 3, line 112) and the Institutional Review Board Statement (Page 12, line 324). Add approval number of the Ethical Committee. 
  • Provide data on the statistical analysis in the separate sub-section.
  • Please consider rephrasing the names of the tables to be more concise.
  • Do not repeat results in the Discussion section (Page 10, lines 238-241).
  • Please add the Conclusion section.

Author Response

Replies to the Reviewer

Reviewer 1

  • The Abstract lacks the conclusion sentence or future directions.

We added conclusion sentence to abstract.

  • Please use the “COVID-19 “abbreviation consistent throughout the whole manuscript.

We unified nomenclature in manuscript as requested.

  • What means “MRI studies…”? (Page 2, line 85)

Now it`s corrected to NMR (Nuclear Magnetic Resonance).

  • The authors should provide more data related to the study design in the Materials and Methods section. Where was a study conducted? When? Add inclusion criteria, followed by exclusion criteria, and then provide data on the methodology.

Thank you for this suggestion. We added information explaining location of the study, inclusion and exclusion criteria and other data describing our methodology in Materials and Methods section.

  • Unify the name of the Ethical Committee in the Material and method section (Page 3, line 112) and the Institutional Review Board Statement (Page 12, line 324). Add approval number of the Ethical Committee. 

We unified the name of the Committee and added the date and number of approval

  • Provide data on the statistical analysis in the separate sub-section.

Thank you for this suggestion. The statistical sub-section was created.

  • Please consider rephrasing the names of the tables to be more concise.

Due to extensive change in statistics, according to other Reviewers we completely changed the display of data. Now names of tables precisely describe its content.

  • Do not repeat results in the Discussion section (Page 10, lines 238-241).

We removed repetition of results from Discussion.

  • Please add the Conclusion section.

We added Conclusion section to the manuscript as suggested by Reviewer.

Reviewer 2 Report

The article by Klimkiewicz and co-workers have aimed in understanding the relationship between cognitive functioning, clinical data, nutrition indexes in patients discharged from the COVID hospital. They measured cognitive impairment using Montreal Cognitive Assessment (MoCa). The analysis showed that the group with lower MoCa results had significantly higher phosphate levels. Additional analysis showed a statistical tendency between the selected nutritional indexes, clinical course, and MoCA scores. Additionally, in this manuscript, the researchers have shown that the short-term memory functions were the most disturbed among examined participants. Their result has been discussed with a potential application in clinical practice. Their data helps in linking short-term cognitive impairment in people having COVID 19 infection history and forms the basis of the existence of COVID Fog.

The article is well written and the use of the English language is understandable. The study has been well designed and meets good technical standards. The methods, tools, software, reagents, and tables are well described and will aid in producing reproducible data by independent groups. The conclusions of this article are important and are of significant relevance as it emphasizes the existence of post-COVID 19 clinical pathologies in patients who have recovered from standard COVID 19 known symptoms. This emphasizes the need for additional diagnostic and pharmacological tools/strategies to be administered on people discharged from hospitals post-COVID 19 generic clinical symptoms recovery. Also, the article highlights a lot of background of the routes of entry of the COVID 19 virus in the nervous systems and their mode of action via neuronal, glial cell, lowering mitochondrial ATP. They also highlight that malnutrition can be one of the potential causes of COVID 19 disease.  Malnutrition can also be pointed to as an environmental susceptible factor leading to COVID 19 in certain individuals and also increasing the severity of the disease.

However, there are some minor points that need to be addressed for the article which might help the article to be widely appreciated amongst a larger group of readers with varied scientific backgrounds.

  1. A table or a chart highlighting the clinical criteria of the patients selected. The table should include, the total number of subjects (Control and COVID19 recovered), Male-Female ratio, Age, clinical symptoms during covid 19 infection, additional genetic predisposition if available for the patients.
  2. The Figure or table heading should be in the beginning. The table or figure title should be followed by the table or figure. The researchers should describe in a few lines the content of the table or the figure, followed by the other details of N= ….and Standard error mean.
  3. All graphs or statistical significance values should be referred to with respect to Standard error mean instead of Standard deviation.
  4. The results of the nutritional status of the control and the COVID 19 patients should be shown in the main manuscript table. So that the readers are able to understand the relevance of the nutritional status in post COVID19 recovery and declining the COVID 19 FOG status in infected individuals.

Author Response

Replies to the Reviewer

Reviewer 2

However, there are some minor points that need to be addressed for the article which might help the article to be widely appreciated amongst a larger group of readers with varied scientific backgrounds.

  1. A table or a chart highlighting the clinical criteria of the patients selected. The table should include, the total number of subjects (Control and COVID19 recovered), Male-Female ratio, Age, clinical symptoms during covid 19 infection, additional genetic predisposition if available for the patients.

Thank you for this comment. As we highlighted in Methods, we included only hospitalized patients recovered form sever COVID-19 just before discharging home. Severity of the disease were described by level of the inflammation state, level of oxygenation and minimal peripheral blood saturation. Male-Female ratio was complemented. However, the genetic predisposition data was not available.

  1. The Figure or table heading should be in the beginning. The table or figure title should be followed by the table or figure. The researchers should describe in a few lines the content of the table or the figure, followed by the other details of N= ….and Standard error mean.

We put the name of the tables ahead of tables.

  1. All graphs or statistical significance values should be referred to with respect to Standard error mean instead of Standard deviation.

Thank you for this comment. However, all figures were removed as not associated with the nutrition.

  1. The results of the nutritional status of the control and the COVID 19 patients should be shown in the main manuscript table. So that the readers can understand the relevance of the nutritional status in post COVID19 recovery and declining the COVID 19 FOG status in infected individuals.

As we changed statistics according to Reviewers’ suggestions, we hope that new tables are more convincing and display results in more transparent way

Reviewer 3 Report

Thank you for providing me the opportunity to review this interesting manuscript, aimed to explore the relationships between neuropsychological measures of the cognitive performance, nutrition indexes and other clinical data. The objective of the study is of clinical interest, and the manuscript seems appropriate for Nutrients readers. However, the study must be reviewed to guarantee journal publication standards. There are some aspects that are not adequately defined, and some sections must be reviewed prior to the decision of being published.

- Abstract. This section is short, in particular the description of the method and the results. Some contents to be included: the number of participants (also the distribution sex and age), origin of the sample, what other indicators were assessed and analyzed (in addition to the MoCa test), the description of the results and possible implications of the empirical results.

- Introduction. This section is well written and structured. However, I consider that it could be improved with these modifications: a) include a new paragraph (previous to the objectives) with studies that relate nutritional status with the affectation severity and progression of COVID-19 (there is a large number of works in the literature); b) write in a separate paragraph the objectives, followed by the empirical hypotheses (deduced from the literature review).

- Results section. Different concerns should be addressed.

  1. Table 1. This table includes some factor scales with “*”, which indicates “aggregated score of specified function tests divided by maximal score of this function”. I do not understand the meaning of this footnote. If these are "new" scales defined by the authors, this information is not appropriate for the study, since these scores have not achieved adequate psychometric indexes in validation studies that guarantee its reliability and validity.
  2. Tables 1-2. Since these tables provide the descriptive for the variables of the study, I suggest to combine in just one table. I also suggest adding the participants’ chronological age in this table.
  3. The analyses provided comparing the two conditions based on the MoCa cut-off have been obtained with very low statistical power because the groups are really too small (sample size equal to 9 and 8, see Table 3). This comprises seriously the internal and external validity of the study. On the other hand, I do not understand why the MoCa sub-tests factor scores have been compared between the groups. I consider that the relationship between the neuropsychological performance and the other clinical measures should be better estimated with correlational models, concretely through a matrix with the partial correlations between each MoCa scores and the remaining variables. Important: due the strong relationship between the variables of the study with the age and the education level (specifically with the neurologic performance), the analyses of the study should be adjusted by these two covariates. Without these adjustment, the crude-evidence obtained in the study are not appropriate since biases due to confounding variables could be impacting these results.
  4. Table 3. This table contains some of the correlational analyses that I suggest in my previous comment. I suggest extending this table with all the neuropsychological measures (for example in columns) and the clinical variables (for example in rows), and include the adjustment by age-education.
  5. Table 3. Correlation coefficients have only been interpreted as relevant for p-values into the significance-range (p<0.05). But due the strong association between the null-hypothesis tests for correlations and the sample size (high correlations usually achieve non-significant results in samples with low sample size, and contrary low correlations achieve significant results in samples with high sample size), these coefficients should be interpreted based on effect size: poor-low for |R|<0.24, mild-moderate for |R|>0.24 and high-good for |R|>0.37. (Thresholds of 0.24 and 0.37 correspond to Cohen’s-d values of 0.50 and 0.80, respectively). P-values are not required for the partial-correlation-matrix table.
  6. Figure 1. This figure does not seem necessary for the study. In fact, a broad dispersion in the scatter-plot is evidenced in this non-adjusted plot.
  7. Figure 2. This figure is not appropriate. The results concerning mean and SD should be provided in the descriptive Table. If authors consider that a figure could contribute for a better understanding of the distributions, the bar-lines plots should be changed by boxplots.
  8. Table 5. I do not understand why the authors obtained the correlational analysis for the neurological variables in the study. No objective is related to this particular analysis. In addiction, it is strange to provide the symmetric matrix with the p-values obtained in the correlational models instead of the correlation estimates. (Note that the symmetry of this table made it not appropriate to include upper and lower part simultaneously). I suggest do not include this analyses, and in the case of including: a) add a specific objective; b) include partial correlations adjusted by age-education.

- Discussion: this section should be reviewed according to the changes in the statistical analysis and the results section.

Author Response

Replies to the Reviewer

Reviewer 3

Thank you for providing me the opportunity to review this interesting manuscript, aimed to explore the relationships between neuropsychological measures of the cognitive performance, nutrition indexes and other clinical data. The objective of the study is of clinical interest, and the manuscript seems appropriate for Nutrients readers. However, the study must be reviewed to guarantee journal publication standards. There are some aspects that are not adequately defined, and some sections must be reviewed prior to the decision of being published.

- Abstract. This section is short, in particular the description of the method and the results. Some contents to be included: the number of participants (also the distribution sex and age), origin of the sample, what other indicators were assessed and analyzed (in addition to the MoCa test), the description of the results and possible implications of the empirical results.

Thank you for this suggestion. We included more information in abstract, including demography of the sample, as well as specific names of lab tests and their correlation with MoCA score. Also potential implication are pictured.

- Introduction. This section is well written and structured. However, I consider that it could be improved with these modifications: a) include a new paragraph (previous to the objectives) with studies that relate nutritional status with the affectation severity and progression of COVID-19 (there is a large number of works in the literature); b) write in a separate paragraph the objectives, followed by the empirical hypotheses (deduced from the literature review).

Thank you for this suggestion. We added references relating nutrition status with clinical course of COVID-19. Also we added hypothesis on the end of “Introduction” section.

Now we state that: ”There is an evidence linking nutrition status with clinical course of COVID-19. James at associates divided field of nutrition into 13 nutrition-related components and their potential interactions with COVID-19. These components are as follows: overweight, obesity, and diabetes; protein-energy malnutrition; anemia; vitamins A, C, D, and E; PUFAs; iron; selenium; zinc; antioxidants; and nutritional support [29]. Nutrition is important player, when comes to susceptibility to infection with SARS-CoV-2 and progression of COVID-19. Diet affects mucosal immune function, immune cells’ function, viral replication and finally inflammation [29]. As nutrition is important for both COVID-19 progression, and cognitive impairment we hypothesize that malnutrition could trigger CF symptoms, and conversely, a proper nutrition is able to ameliorate cognitive dysfunction after COVID-19. As malnutrition is widespread among both surgical and medical patients [30,31] we found purposive to investigate potential correlation between nutrition state and CF.”

- Results section. Different concerns should be addressed.

  1. Table 1. This table includes some factor scales with “*”, which indicates “aggregated score of specified function tests divided by maximal score of this function”. I do not understand the meaning of this footnote. If these are "new" scales defined by the authors, this information is not appropriate for the study, since these scores have not achieved adequate psychometric indexes in validation studies that guarantee its reliability and validity.

Thank you for this question. Indeed, the “aggregated score” name could be confounding. “For cognitive functions estimated in more than one test, the sum of specific results for the given cognitive ability tests was divided by the sum of their maximal scores to achieve a decimal fraction ranging from 0,00 to 1,00, which was considered for statistics. Visuospatial function expresses the results of two tests: the drawing figure and joining points test and the clock drawing test. Attention function consisted of attention digits, letters and subtraction tests. Language function express the results of the repetition and the fluency tests.” To clarify that calculation, we completed methods accordingly. No new scales were created.

  1. Tables 1-2. Since these tables provide the descriptive for the variables of the study, I suggest to combine in just one table. I also suggest adding the participants’ chronological age in this table.

Thank you for this valuable comment. We combined Tables 1-2 into one table. We present further statistical results as adjusted to age. Thus, we suppose that presentation of the individual age of each participant is not necessary.

  1. The analyses provided comparing the two conditions based on the MoCa cut-off have been obtained with very low statistical power because the groups are really too small (sample size equal to 9 and 8, see Table 3). This comprises seriously the internal and external validity of the study. On the other hand, I do not understand why the MoCa sub-tests factor scores have been compared between the groups.

Thank you for this comment. We agree with this concern and removed this data.

I consider that the relationship between the neuropsychological performance and the other clinical measures should be better estimated with correlational models, concretely through a matrix with the partial correlations between each MoCa scores and the remaining variables. Important: due the strong relationship between the variables of the study with the age and the education level (specifically with the neurologic performance), the analyses of the study should be adjusted by these two covariates. Without these adjustment, the crude-evidence obtained in the study are not appropriate since biases due to confounding variables could be impacting these results.

Thank you for this very valuable comment. We recalculated data, and present partial correlations between each MoCA scores and the remaining variables, adjusted to age and the education level.

  1. Table 3. This table contains some of the correlational analyses that I suggest in my previous comment. I suggest extending this table with all the neuropsychological measures (for example in columns) and the clinical variables (for example in rows) and include the adjustment by age-education.

Thank you for this comment. Table 3 was replaced with partial correlations adjusted to age and the education level.

  1. Table 3. Correlation coefficients have only been interpreted as relevant for p-values into the significance-range (p<0.05). But due the strong association between the null-hypothesis tests for correlations and the sample size (high correlations usually achieve non-significant results in samples with low sample size, and contrary low correlations achieve significant results in samples with high sample size), these coefficients should be interpreted based on effect size: poor-low for |R|<0.24, mild-moderate for |R|>0.24 and high-good for |R|>0.37. (Thresholds of 0.24 and 0.37 correspond to Cohen’s-d values of 0.50 and 0.80, respectively). P-values are not required for the partial-correlation-matrix table.

Thank you for this comment. Table 3 was replaced with partial correlations without p-value. The effect-size interpretation we completed in the Methods section.

  1. Figure 1. This figure does not seem necessary for the study. In fact, a broad dispersion in the scatter-plot is evidenced in this non-adjusted plot.

Thank you for this comment. Figure 1 was removed as not associated with the nutrition.

  1. Figure 2. This figure is not appropriate. The results concerning mean and SD should be provided in the descriptive Table. If authors consider that a figure could contribute for a better understanding of the distributions, the bar-lines plots should be changed by boxplots.

Thank you for this comment. Figure 2. was removed as not associated with the nutrition.

  1. Table 5. I do not understand why the authors obtained the correlational analysis for the neurological variables in the study. No objective is related to this particular analysis. In addiction, it is strange to provide the symmetric matrix with the p-values obtained in the correlational models instead of the correlation estimates. (Note that the symmetry of this table made it not appropriate to include upper and lower part simultaneously). I suggest do not include this analyses, and in the case of including: a) add a specific objective; b) include partial correlations adjusted by age-education.

Thank you. We agree with this suggestion, and we decided to remove this data.

Reviewer 4 Report

The Manuscript (nutrients-1624748, title: Analysis of the relationship between cognitive impairment, nutritional indexes and the clinical course among COVID patients discharged from hospital- preliminary report) evaluated an interesting topic in COVID patients. The aim of the study was to analyse the relationship between cognitive function, clinical data, and nutrition indexes in patients discharged from the COVID hospital.

The Manuscript was well-written and interesting but showed number of two severe limitations.

First, in this study were evaluated only patients with COVID-19, however a control group was necessary to compare nutritional status. Second, the low number of patients enrolled (9+8).

Authors should increase the sample size or Authors should highlight the preliminary character of the work in the title. In addition, power calculations considering a critical effect size are required to ensure the study was appropriately powered.

Minor comments:

Line 24 It should be “…and nutrition indexes”.

Line 40-41 Authors should considere also the presence of phantosmia and phantogeusia in patiens as a long sequele of COVID-19 (Ercoli et al., 2021 https://doi.org/10.1007/s10072-021-05611-6)

Line 134-137 Authors should explain the criteria for the evaluation of nutritional status. Why did not evaluate for example potassium or calcium concentration?

Lines 158, 222, 224, 225, 232, 238, 239, 240 It should be “MoCA”.

Author Response

Replies to the Reviewer

Reviewer 4

The Manuscript (nutrients-1624748, title: Analysis of the relationship between cognitive impairment, nutritional indexes, and the clinical course among COVID patients discharged from hospital- preliminary report) evaluated an interesting topic in COVID patients. The aim of the study was to analyze the relationship between cognitive function, clinical data, and nutrition indexes in patients discharged from the COVID hospital.

The Manuscript was well-written and interesting but showed number of two severe limitations.

First, in this study were evaluated only patients with COVID-19, however a control group was necessary to compare nutritional status. Second, the low number of patients enrolled (9+8).

Authors should increase the sample size or Authors should highlight the preliminary character of the work in the title.

Thank you for this comment. We clarified the information for potential readers that this is preliminary study. The title comprises this information now. Also, we expanded statistical section with explanation that we calculated minimal sample size and our sample, though limited, is enough.

In addition, power calculations considering a critical effect size are required to ensure the study was appropriately powered.

Thank you for this valuable remark. We completed statistics with sample size calculation: “For the sample size calculation analysis of correlation between short-term memory score and albumin concentration (r=0.772; p=0.072) was performed in the group of 10 initially included patients. To achieve the power of 0.90 and p<0.05, minimal sample size was 12”; and Results section with the power analysis: “Albumin concentration correlated significantly with short-term memory (r=0.712; p=0.021; power of the test 0.87).”

Minor comments:

Line 24 It should be “…and nutrition indexes”.

The text was corrected according to Reviewer suggestion.

Line 40-41 Authors should considere also the presence of phantosmia and phantogeusia in patiens as a long sequele of COVID-19 (Ercoli et al., 2021 https://doi.org/10.1007/s10072-021-05611-6)

We added those symptoms to the list observed in long COVID as well as suggested literature supporting this to reference list.

Line 134-137 Authors should explain the criteria for the evaluation of nutritional status. Why did not evaluate for example potassium or calcium concentration?

Thank you for this valuable comment. We added sentence explaining why we chose those particular markers and added calcium to analysis.

Lines 158, 222, 224, 225, 232, 238, 239, 240 It should be “MoCA”.

We looked trough manuscript and unified abbreviation into “MoCA”

Reviewer 5 Report

The study is interesting (considering the pandemic generated by covid-19). But the introduction, methodology, results and discussion are very far from nutrition. This is a major weakness of the manuscript. In pubmed, it is possible to find many studies on nutritional status in patients with covid-19. I suggest that the manuscript be oriented to nutrition.

I. Major Comments
1. In the introduction the authors do not refer to nutritional indices. This is a major problem in the manuscript. I suggest including a paragraph on this point, and reducing or transferring to the discussion other aspects that are presented in the introduction.

2. The authors present a manuscript that mainly addresses clinical aspects related to covid-19, but the nutritional points are very limited (methodology, results and especially the discussion).

3. The study presents clinical aspects that could be correlated with indicators of nutritional status (in plasma or anthropometric).

4. Clinical studies that consider nutritional status present and/or discuss results related to nutrients and/or the metabolic status of the patient, and some clinical condition. In this study those analyzes are very limited.

5. Aspects that should be included in this study:
- Blood lipids (Triglycerides, HDL, LDL and VLDL-cholesterol). This allows us to analyze what can happen with cardiovascular health
- Erythrocytes and homoglibin in blood. Important role of iron.
- Vitamins and immune system (vitamin D and C)

6. The absence of anthropometric parameters (weight, height and BMI) is worrying. This analysis would allow evaluating what happens to body weight and potentially body composition.

7. The authors do not discuss anything about diet, nutrients and covid-19.

8. How could nutrients be related to the cognitive impairment that would be observed in patients who had the disease caused by covid-19?
8.1. What nutrients could be related to a lower or higher inflammation and especially neuro-inflammation? Zinc, selenium, proteins, vitamin E or C.
8.2. Would it be possible to establish one or more hypotheses about how specific nutrients could influence a greater or lesser congenital deterioration?
8.3. Could nutrients such as DHA or vitamin D in patients with covid-19 prevent or attenuate cognitive deterioration?

9. Finally, it was very difficult for me to relate the study to nutritional indices.

10. How can the results of Figure 1 or Figure 2 be related to the nutritional indicators?

II. Minor comments:

1. Improve the wording of the objective of the study
2. In the title I suggest replacing covid with covid-19
3. Define SARS-CoV-2 in the manuscript

Author Response

Replies to the Reviewer

Reviewer 5

The study is interesting (considering the pandemic generated by covid-19). But the introduction, methodology, results and discussion are very far from nutrition. This is a major weakness of the manuscript. In PubMed, it is possible to find many studies on nutritional status in patients with covid-19. I suggest that the manuscript be oriented to nutrition.

  1. Major Comments
    1. In the introduction the authors do not refer to nutritional indices. This is a major problem in the manuscript. I suggest including a paragraph on this point and reducing or transferring to the discussion other aspects that are presented in the introduction.

Thank you for this comment. We added paragraph about nutrition and covid fog to introduction.

  1. The authors present a manuscript that mainly addresses clinical aspects related to covid-19, but the nutritional points are very limited (methodology, results and especially the discussion).

According to suggestions we added more parameters to analysis, so now the paper is more oriented towards nutrition.

  1. The study presents clinical aspects that could be correlated with indicators of nutritional status (in plasma or anthropometric).

Thank you for this remark. We fully agree with reviewer in this point. From methodological and scientific point of view this is correct. However, this study was performed under condition of temporary hospital, understaffed, and basing on “damage control” rule. So, lab tests were available, as used to monitor treatment. Anthropometric date was approximated basing on patients’ statement (last time I checked I was of 80 kilograms and of 179 cm of height). We were unable to monitor patient`s weight.

  1. Clinical studies that consider nutritional status present and/or discuss results related to nutrients and/or the metabolic status of the patient, and some clinical condition. In this study those analyzes are very limited.

We have added paragraph in discussion comparing our results to other studies. As we added more markers to analysis also discussion is expanded.

  1. Aspects that should be included in this study:
    - Blood lipids (Triglycerides, HDL, LDL and VLDL-cholesterol). This allows us to analyze what can happen with cardiovascular health

We expanded analysis and included HDL, LDL, triglycerides according to above suggestion.
- Erythrocytes and hemoglobin in blood. Important role of iron.

We added hemoglobin and RBC to analysis according to Reviewer`s comment.
- Vitamins and immune system (vitamin D and C)

Thank you for this suggestion. We completed nutritional parameters with HDL, LDL, triglycerides, erythrocytes, and hemoglobin. However, data concerning VLDL, and vitamins were not available.

  1. The absence of anthropometric parameters (weight, height and BMI) is worrying. This analysis would allow evaluating what happens to body weight and potentially body composition.

We fully agree with reviewer in this point. From methodological and scientific point of view this is correct. However, this study was performed under condition of temporary hospital, understaffed, and basing on “damage control” rule. So, laboratory tests were available, as used to monitor treatment. Anthropometric data was approximated basing on patients’ statement (“last time I checked I was of 80 kilograms and of 179 cm of height”). We were unable to monitor patient`s weight. Because of this, we decided not to include anthropometric data, as they carry a high risk for bias/mistake.  

  1. The authors do not discuss anything about diet, nutrients and covid-19.

According to this valuable comment we expanded discussion section, also in introduction we added paragraph linking nutrition status to COVID-19 progression.

  1. How could nutrients be related to the cognitive impairment that would be observed in patients who had the disease caused by covid-19?

We added paragraph in discussion, with possible explanation of how nutrition status can interfere with severity of cognitive decline

8.1. What nutrients could be related to a lower or higher inflammation and especially neuro-inflammation? Zinc, selenium, proteins, vitamin E or C.

Unfortunately, we were unable to include markers of neuroinflammation, zinc, selenium, vitamins into analysis. However, this is very interesting and could be of potential use for future studies.

8.2. Would it be possible to establish one or more hypotheses about how specific nutrients could influence a greater or lesser congenital deterioration?

Thank you. We added hypothesis to our study. Now we are stating that:

“There is evidence linking nutrition status with clinical course of COVID-19. James at associates divided field of nutrition into 13 nutrition-related components and their potential interactions with COVID-19. These components are as follows: overweight, obesity, and diabetes; protein-energy malnutrition; anemia; vitamins A, C, D, and E; PUFAs; iron; selenium; zinc; antioxidants; and nutritional support [29]. Nutrition is important player, when comes to susceptibility to infection with SARS-CoV-2 and progression of COVID-19. Diet affects on mucosal immune function, immune cells’ function, viral replication and finally inflammation [29]. As nutrition is important for both COVID-19 progression, and cognitive impairment we hypothesize that malnutrition could trigger CF symptoms, and conversely, a proper nutrition is able to ameliorate cognitive dysfunction after COVID-19. As malnutrition is widespread among both surgical and medical patients [30,31] we found purposive to investigate potential correlation between nutrition state and CF.”

8.3. Could nutrients such as DHA or vitamin D in patients with covid-19 prevent or attenuate cognitive deterioration?

  1. Finally, it was very difficult for me to relate the study to nutritional indices.

As we expanded analysis about more nutrition indexes, according to Reviewers we find it now related with nutrition.

  1. How can the results of Figure 1 or Figure 2 be related to the nutritional indicators?

Thank you for this comment. We agree with it. Thus figures 1-2 were removed as not associated with the nutrition.

  1. Minor comments:
  2. Improve the wording of the objective of the study

 We corrected this section.

  1. In the title I suggest replacing covid with covid-19

We corrected naming into COVID-19 consistently in whole text.

  1. Define SARS-CoV-2 in the manuscript

We explained abbreviation (severe acute respiratory syndrome coronavirus 2).

Round 2

Reviewer 3 Report

Thank you for providing me the opportunity to review the new version of this interesting manuscript. The authors have made all the changes I had suggested in my previous review.

Author Response

Thank you for your valuable comments.

Reviewer 4 Report

I appreciate the revised draft of the Manuscript (nutrients-1624748). Authors revised the article in line to Reviewer's suggestion.

Author Response

Thank you for your valuable comments.

Reviewer 5 Report

The authors answered most of my questions or suggestions. It would be ideal to include more clinical history of the patients. But the authors' response is understandable.

Author Response

Thank you for your valuable comments.